# Recovery of Bioactive Compounds from Strawberry (*Fragaria* × *ananassa*) Pomace by Conventional and Pressurized Liquid Extraction and Assessment Their Bioactivity in Human Cell Cultures

**DOI:** 10.3390/foods10081780

**Published:** 2021-07-31

**Authors:** Milda Pukalskienė, Audrius Pukalskas, Lijana Dienaitė, Simona Revinytė, Carolina V. Pereira, Ana A. Matias, Petras Rimantas Venskutonis

**Affiliations:** 1Department of Food Science and Technology, Kaunas University of Technology, Radvilenu Rd. 19, LT-50254 Kaunas, Lithuania; milda.skemaite@ktu.lt (M.P.); audrius.pukalskas@ktu.lt (A.P.); lijana.dienaite@gmail.com (L.D.); simonar91@gmail.com (S.R.); 2IBET—Instituto de Biologia Experimental e Tecnológica, Food & Health Division Apartado 12, 2780-901 Oeiras, Portugal; carolina.pereira@ibet.pt (C.V.P.); amatias@ibet.pt (A.A.M.)

**Keywords:** strawberry pomace, phytochemical composition, extraction, antiproliferative activity, cytotoxicity

## Abstract

Pressing strawberries for juice generates large amounts of pomace, containing valuable nutrients and therefore requiring more systematic studies for their valorization. This study compared conventional solid-liquid (SLE) and pressurized liquid (PLE) extractions with ethanol (EtOH) and H_2_O for the recovery of bioactive compounds from strawberry pomace. The composition and bioactivities of the products obtained were evaluated. Among 15 identified compounds, quercetin-3-glucuronide, kaempferol-3-glucuronide, tiliroside, ellagic, malic, succinic, citric and *p*-coumaric acids were the most abundant constituents in strawberry pomace extracts. SLE-EtOH and PLE-H_2_O extracts possessed strong antioxidant capacity in DPPH^•^ and ABTS^•+^ scavenging and oxygen radical absorbance capacity (ORAC) assays. Cytotoxicity, antiproliferative and cellular antioxidant activities in human cells of PLE-EtOH and PLE-H_2_O extracts were also evaluated. PLE-EtOH and PLE-H_2_O extracts possessed strong antioxidant activity, protecting Caco-2 cells upon stress stimuli, while PLE-EtOH extract showed higher antiproliferative activity with no cytotoxicity associated. In general, the results obtained revealed that properly selected biorefining schemes enable obtaining from strawberry pomace high nutritional value functional ingredients for foods and nutraceuticals.

## 1. Introduction

Strawberries are one of the most popular berries in the world with a global annual production of 9.22 MT. The formation of the cultivated strawberry (*Fragaria × ananassa* Duchesne) started in the eighteenth century when strawberry culture became increasingly limited to the clones of this hybrid species. The genus *Fragaria* belongs to the one of the most important economically Rosaceae family. *Fragaria* is a member of the subfamily Rosoideae and consists of approximately 20 diploid, tetraploid, hexaploid and octoploid species [1].

Freshly harvested strawberries are highly perishable fruits due to fast post-harvest decay, high respiration rate and environmental stress. Shelf life of fresh strawberries is approx. 2–3 days at room temperature [2]. Therefore, a large fraction of harvested strawberries are processed into numerous products such as jams, purees, wine, juice and others. Processing of strawberries in some cases generates by-products; for instance, the residues (pomace) in juice production constitute approximately 4–11% of fruit weight. Currently, a large part of such by-products are used very inefficiently, e.g., for composting or animal feeding. It is well documented that strawberry pomace, consisting of seeds, stalks and pulp, contains valuable nutrients such as phenolic compounds (anthocyanins, proanthocyanidins, ellagic and other phenolic acids, ellagitannins), minerals, dietary fiber and others [3,4]. For instance, the amount of hydrolysable ellagitannins in strawberries, depending on the origin and ripeness of fruits, may reach 637 mg/kg fresh weight on average [5].

Due to the presence of a significant amount of biologically active substances and high nutritional value, strawberry pomace has a great potential to be used as a source for health beneficial ingredients for functional foods, nutraceuticals, cosmeceuticals and other healthy natural products [6,7]. Several studies reported antiproliferative activities of strawberry extracts, while pomace products have also been considered as a possible preventive means against various diseases, such as cardiovascular disorders, cancer and atherosclerosis [8,9,10]. For instance, McDougal et al. [11] suggested that polyphenol-rich strawberry extract might be useful for mitigating diabetes via inhibition of α-glucosidase enzyme and reducing the postprandial absorption of glucose, which is produced in the small intestine via breaking the starch and disaccharides. Kosmala et al. [12] determined that aqueous and aqueous/alcoholic extracts of strawberry pomace showed similar effects on the enzymatic activity in gastrointestinal tract, which is obtained by substituting dietary cellulose with more easily fermentable fructooligosaccharides (FOS). Some studies reported in vivo effects of strawberry pomace products as well. Polyphenol-rich and depleted of phenolic compounds strawberry pomace added to the diet of rats had similar positive effects on gastrointestinal, blood and tissue biomarkers of experimental animals by reducing metabolic complications [13]. Later, Juskiewicz et al. [5], reported that acetone extract of strawberry pomace lowered lipaemia and glycemia indicators in Wistar rats. However, the data on the bioactivities of strawberry pomace extracts are rather scarce, particularly using human cells and physiologically important enzymes.

More systematic and comprehensive studies are also required for valorizing strawberry pomace as a source of various functional ingredients. Biorefining of strawberry pomace into several fractions containing various classes of nutrients (including bioactive compounds), possessing different activities and physical properties is a promising approach. The advantages of such an approach have been recently demonstrated for multistep processing of chokeberry [14,15], cranberry [16], guelder-rose berry [17] and raspberry pomace [18]. Several nutritionally valuable fractions were obtained from each type of pomace achieving a sustainable ‘zero waste’ processing task. Moreover, promising results were obtained in terms of shifting to green chemistry-based high-pressure extraction/fractionation systems using supercritical carbon dioxide (SFE-CO_2_), pressurized ethanol and water (PLE). In general, high-pressure-based extractions have been proved to be more efficient than conventional processes such as extract yield, time of extraction, solvent-free extracts (in SFE-CO_2_) and reduced consumption of solvent (in PLE). For instance, PLE, which uses solvents in their subcritical state, improves the solubility of phytochemicals and their transfer from the solid matrix in a shorter time both using organic solvents and water [19]. However, the composition and distribution of phenolic compounds and other constituents in the pomace derived products highly depend on plant origin; therefore, extraction schemes and conditions should be properly designed and investigated individually for each type of pomace.

The aim of this study was to compare conventional and pressurized liquid extraction methods in developing integrated biorefining schemes for recovery of bioactive compounds and other valuable nutrients from strawberry pomace and to evaluate the extracts obtained by using antioxidant, cytotoxicity and antiproliferative activity assays using human HT 29 and Caco-2 cells, as well as by screening their phytochemical composition by chromatographic and spectroscopic methods. It is expected that such results will provide essential data for valorizing strawberry pomace in the development of various functional ingredients for foods and nutraceuticals.

## 2. Materials and Methods

### 2.1. Chemicals, Cell Cultures and Other Materials

Trolox, 97% (6-hydroxy-2,5,7,8-tetramethylchroman-2-carboxylic acid); ABTS (2,2′-azino-bis(3-ethylbenzthiazoline-6-sulphonic acid); DDPH^•^ (1,1-diphenyl-2- picrylhydrazyl stable radical); gallic acid 99% (3,4,5-trihydroxybenzoic acid); 2,2′-azobis-2-methyl-propanimidamide dihydro-chloride (AAPH); microcrystalline cellulose (20 μm); Na_2_CO_3_, cyanidin-3-glucoside (>95%) were from Sigma-Aldrich (Steinheim, Germany); 2-(3-hydroxy-6-oxo-xanthen-9-yl)benzoic acid (Fluorescein (FL), Folin and Ciocalteu’s phenol reagent (2M) from Fluka Analytical, Bornem, Belgium); NaCl, KCl, KH_2_PO_4_, K_2_S_2_O_8_ (Lach-Ner, Brno, Czech Republic), Na_2_HPO_4_ (Merck KGaA, Darmstadt, Germany), ASE filters (Glass Fiber-(X)-Cellulose, Dionex Corporation, Sunnyvale, CA, USA), diatomaceous earth (100% SiO_2_, Dionex Corporation, Sunnyvale, CA, USA), ethanol (EtOH, 96.3%, agricultural origin, Stumbras, Kaunas, Lithuania), liquid nitrogen (AGA SIA, Riga, Latvia), CO_2_ and N_2_ (99.9%, Gaschema, Jonava region, Lithuania). All other solvents used for extraction were of analytical grade; for MS analysis was MS-grade. Human Caco-2 and HT29 cell lines were obtained from Deutsche Sammlung von Mikroorganismen und Zellkulturen (DSMZ, Germany) and American Type Culture Collection (ATCC, USA), respectively. Cell culture medium and supplements were purchased from Invitrogen (Gibco, Invitrogen Corporation, UK). Phosphate Buffered Saline (PBS) for cells was from (Sigma-Aldrich, St. Louis, MO, USA); CellTiter 96^®^ AQueous One Solution Cell Proliferation Assay was from (Promega, Madison, WI, USA); quercetin, DMSO (dimethyl sulfoxide), DCFH-DA (2′,7′-dichlorofluorescin diacetate) from (St. Quentin Fallavier, France).

### 2.2. Preparation of Berry Pomace and Determination of Its Proximate Composition

Strawberry pomace (*Fragaria × ananassa*) was kindly donated by the company “Anykščių vynas” (Anykščiai, Lithuania) in 2017 immediately after juice pressing. The pomace, containing pulp, stalk and seeds was air-dried in a SENCOR convection dryer at 40 °C for 48 h. After drying the material was ground in a ZM 200 ultra-centrifugal mill, using 1 mm sieve (Retsch, Haan, Germany) at 12,000 rpm. Ground material was stored in tightly closed, dry glass jars in a dark, well-ventilated place. The dried sample of strawberry pomace, which was used in this study, is stored at 4 °C in the Department of Food Science and Technology (no. SP-2017) and is available upon request.

Moisture content was determined by drying at 104 °C to constant mass. The content of ash was determined after incineration in a muffle furnace at 550 °C for 3 h. The crude protein content was measured by the Kjeldahl procedure using nitrogen conversion factor 5.3 (AOAC, 950.09) [20], the crude lipid content was determined by a Soxhlet method (AOAC, 963.15) [20] using hexane. All analyses were performed in triplicate, and the results were expressed as grams per 100 g of dry matter of strawberry pomace.

### 2.3. Extraction of Polar Constituents from Strawberry Pomace

Pressurized liquid extraction (PLE). PLE was performed with ethanol (EtOH) and water (H_2_O) in an accelerated solvent extractor ASE 350 (Dionex, Sunnyvale, CA, USA) equipped with a solvent-controlling unit. Extractions were performed in 65 mL cells at 10.3 MPa and at 90 °C, and 110 °C for EtOH and H_2_O, respectively. Ground strawberry pomace (5 ± 0.001 g) was loaded into the cell with 3 ± 0.001 g of diatomaceous earth above and below the sample to avoid any void spaces, and with two cellulose filters in both ends to avoid particle leak to the system. Then, the cell was placed into the carrousel to start an automatic extraction sequence; it was heated 5 min to the pre-set extraction temperature, ad pressurized for 15 min (3 cycles × 15 min). The total volume of solvent was 120 mL. The EtOH was removed in a rotary evaporator Rotavapor R-114 (Büchi Flavil, Switzerland) under vacuum (0.06 MPa) at 40 °C, aqueous solution was freeze-dried (Maxi Dry Lyo, Jonan Allerod, Denmark). All obtained extracts were stored at −20 °C until analysis.

Solid-liquid extraction (maceration) (SLE). SLE was performed for comparing it with PLE. Ground pomace (30 g) was extracted with 150 mL of EtOH or H_2_O for 24 h at room temperature under orbital shaking at 250 rpm. Afterwards, the contents were centrifuged at 6000 rpm for 10 min and filtered. The solvents were removed as in PLE.

### 2.4. Evaluation of Antioxidant Properties of Extracts and Solid Materials

Total phenolic content (TPC). Folin–Ciocalteu method was applied with minor modifications [21]. Briefly, 150 µL of extract (0.5–2.5 mg/mL) or corresponding solvent methanol (MeOH) or H_2_O as a blank was mixed with 750 µL Folin–Ciocalteu’s reagent (2 M), previously diluted with distilled water (1:9, *v*/*v*) and after 3 min 600 μL of 7.5% (*w*/*v*) Na_2_CO_3_ was added. The mixture was kept at 25 °C 2 h in the dark and the absorbance measured in 1 cm path length disposable cuvettes (Greiner Labortech, Alpher a/d Rijn, The Netherlands) at 760 nm in a Genesys 8 UV spectrophotometer (Thermo Spectronic, Rochester, NY, USA). TPC was determined from GA calibration curve (0.025–0.5 mg/mL) and expressed as mg of GA equivalents (GAE) per g of extract.

The ABTS^•+^ scavenging capacity. The assay was performed according to the method of Re at al. [22]. The ABTS^•+^ was produced by reacting 75 mM ABTS in PBS (pH 7.4) with 200 µL K_2_S_2_O_8_ (70 mmol/L) and allowing the mixture to stand in the dark at room temperature for 15–16 h for color development. The working solution of ABTS^•+^ was prepared daily by diluting its stock solution in PBS to reach an absorbance value of 0.70 ± 0.20 at 734 nm. Strawberry pomace extracts were dissolved in EtOH and H_2_O, and diluted in PBS to 0.5–2.5 mg/mL concentration. The aliquots of 25 μL of each extract were added to 1500 μL of ABTS^•+^ solution and the absorbance was read after 2 h at 734 nm in a Genesys 8 UV spectrophotometer. PBS was used as a blank. Trolox Equivalent Antioxidant Capacity (TEAC) values were determined from a Trolox calibration curve built by using 80–1500 μM solutions and the results were expressed as mg of TE per g of extract (mg TE/g) and further recalculated to g of material dry weight (mg TE/g DW).

DPPH^•^ scavenging capacity. The assay was performed by the method of Brand Williams et al. [23]. One thousand μL of freshly prepared MeOH solution, containing 250 μM DPPH^•^, was added to 500 μL of diluted to 0.5–2.5 mg/mL extracts. The absorbance was measured after 2 h of incubation in the dark at room temperature in a Genesys 8 UV spectrophotometer at 517 nm. MeOH was used as a blank. The values were determined from a calibration curve prepared with 3–100 μM solutions of Trolox and expressed as mg TE/g extract.

Oxygen Radical Absorbance Capacity (ORAC). The assay was performed as described by Prior et al. [24] with minor modifications. A multiple-detection microplate FLUOstar Omega reader (BMG Labtech, Offenburg, Germany) with fluorescent filters (excitation wavelength, 485 nm; emission wavelength, 520 nm) was used. Twenty-five µL of diluted to 0.15–0.75 mg/mL extract or pure MeOH or H_2_O (used as blanks), was mixed with 150 µL of fluorescence probe fluorescein solution (14 µmol/L), preincubated for 15 min at 37 °C followed by the rapid addition of 25 µL of peroxyl radical generator AAPH (240 mmol/L). The fluorescence was recorded every cycle (1 min × 1.1), in total 120 cycles. Trolox was used as a reference antioxidant. Final results were calculated on the basis of the difference in the area under the fluorescein decay curve between the blank and each sample. The ORAC values were determined from its ability to protect the fluorescence of the indicator in the presence of peroxyl radicals.

Antioxidant activity assessment of solid material by the QUENCHER method. TPC and ABTS^•+^ scavenging of raw plant material and solid residues after extractions were determined by using QUENCHER procedure [25]. Solid dilutions were performed by mixing 10 mg of the sample with microcrystalline cellulose, which was also used as a blank. Further experimental procedures were the same as reported for extracts. In ABTS^•+^ assay solid dilutions were performed by mixing 10 mg of sample with microcrystalline cellulose, which was also used as a blank. Further material was diluted with 2 mL ABTS^•+^ (prepared as reported for the extracts), the mixture was vortexed for 120 min to facilitate the surface reaction, centrifuged at 4500 rpm for 5 min, and the absorbance of the supernatant was measured at 734 nm. In TPC assay 10 mg of the sample were mixed with microcrystalline cellulose and with 1.5 mL Folin–Ciocalteu’s reagent solution (1:9). The reagents were mixed, kept for 10 min, neutralized with 1.2 mL of 7.5% sodium carbonate, vortexed for 120 min and centrifuged at 14,000 rpm for 5 min. The absorbance was measured at 765 nm. TPC and TEAC values were expressed as mg GAE/g DW and mg TE/g DW of pomace, respectively. All experiments were replicated four times.

### 2.5. Preparation of Cell Culture and Cellular Assays

Caco-2 cells were cultivated in RPMI-1640 medium supplemented with 10% of heat-inactivated fetal bovine serum and 1% penincilin-streptomycin at 37 °C with 5% CO_2_ in a humidified incubator and routinely grown as a monolayer in 75 cm^2^ culture flasks. Strawberry pomace extracts were dissolved in DMSO and EtOH to a final concentration of 100 mg/mL. The prepared samples were stored at −20 °C in the dark. Cell-based assays were performed using a maximum concentration of solvent, namely 1% and 5% for DMSO and EtOH, respectively.

Cytotoxicity assay in Caco-2 cell monolayer. Cytotoxicity was assessed as previously described by Silva et al. [26]. Caco-2 cells in growth medium were placed in 96-well plates at a density of 2 × 10^4^ cells/well. After 7 days (during this period the cells were renewed every 48 h) the growth medium was removed and replaced with media containing different concentrations of strawberry pomace extracts. Control wells contained growth medium with no extract. After 24 h of incubation at 37 °C, the cells were washed twice with PBS and cell viability was determined using MTS reagent according to manufacturer’s instructions. Reduction in absorbance was measured at 490 nm using a Spark^®^ 10M Multimode Microplate Reader (Tecan Trading AG, Männedorf, Switzerland) and cell viability was expressed in terms of percentage of living cells relative to the control. Experiments were performed in triplicate.

Antiproliferative assay in HT29 cell monolayer. Antiproliferative effect of extracts and standard compounds was evaluated in HT29 cells, as described elsewhere [27]. Briefly, the cells were placed in each well of a 96-well plate at a density of 1 × 10^4^ cells/well. After 24 h the cells were incubated with different concentrations of the samples diluted in culture medium. Control wells contained growth medium with no extract. Cell proliferation was measured after 24 h using MTS reagent, as explained above. The results were expressed in terms of percentage of living cells relative to the control. A minimum of three replicates for each sample was used to determine the antiproliferative activity.

Cellular antioxidant activity (CAA) assay. The CAA of strawberry pomace extracts was assessed using the method of Wolfe and Liu [28]. Caco-2 cells were seeded in growth medium at a density of 2 × 10^4^ cell/well in a 96-well microplate. After 6 days the medium was removed and cells were washed twice with pre-warmed to 37 °C PBS. Afterwards, the cells were treated with 50 µL of PBS/sample/standard (quercetin, 2.5–20 µM) solution and 50 µL of DCFH-DA solution (50 µM) were added and incubated for 1 h at 37 °C and 5% CO_2_. Next, 100 μL of AAPH (12 mM) solution were added to each well containing PBS/quercetin standards/samples, while 100 µL of PBS were added to the blank wells. Finally, the 96-well microplate was placed into a Microplate Fluorimeter FLx800 (Biotek Instruments, VT, USA). The emission wavelength at 540 nm was measured after excitation at 485 nm every 5 min for 1 h. CAA values were expressed as µM of quercetin equivalents per g of extract (µM QE/g). Independent experiments were performed in triplicates.

### 2.6. Identification of Bioactive Components Using the Ultra-Performance Liquid Chromatography—Mass Spectrometry (UPLC-MS)

Non-targeted analysis by UHR-Q-TOF-MS. Phytochemicals were identified by non-targeted screening based on high accuracy mass spectra. An Acquity UPLC system (Waters, Milford, USA) was equipped with a Bruker maXis quadrupole time-of-flight mass spectrometer (UHR-Q-TOF-MS) (Bruker Daltonics, Bremen, Germany). An Acquity BEH C18 column; 1.7 µm, 100 × 2.1 mm, i.d. (Waters, Milford, USA) was used for separation, column temperature was maintained at 40 °C. The gradient elution programmed for mobile phases A (1% formic acid) and B (acetonitrile) was as follows: 0 min, 95% A; 1–3 min, 95–85% A; 3–7 min, 85–50% A; 7–10 min, 50–0% B; 10–12 min, 0% A; 12–14 min, 95% A. The flow rate was 0.4 mL/min, the injection volume 2 μL. An electrospray ionization (ESI) source was used, the spectra were recorded in the mass range of *m*/*z* 100–1500 in the negative mode the capillary voltage was adjusted to +4000 V. The nebulizer pressure was 2.0 bar and the nitrogen flow rate was 10 L/min. For fragmentation study, a data-dependent scan was performed by deploying collision-induced dissociation (CID) using nitrogen as a collision gas at 30 eV. Fullscan and auto MS/MS were set for scanning data at 2 Hz acquisition speed. Data acquisition, handling and instrument control were performed using Compass 1.3 (HyStar 3.2 SR2) software. The phytochemicals were identified by searching the ChemSpider database based on molecular formulas, calculated from accurate mass-to-charge ration matching of the MS data.

Anthocyanins were quantified using cyanidin-3-glucoside as the external standard [15]. Standard stock solution was prepared in MeOH and subsequently diluted to working concentrations. The amounts of individual compounds were expressed as mg/100 g of strawberry pomace DW.

Quantitative analysis of organic acids by UHR-Q-TOF-MS. Quantitative analysis of organic acids was performed using the same UPLC-MS system and method as described above. Malic, citric, quinic and succinic acids were quantified by integrating extracted ion chromatograms of the *m*/*z* values, corresponding to each acid, namely *m*/*z* 133.0131 for malic acid, *m*/*z* 191.0186 for citric acid, *m*/*z* 191.0550 for quinic acid and *m*/*z* 117.0182 for succinic acid. The *m*/*z* values were extracted with an accuracy of 0.02. Standard compounds were chromatographed in the concentration range from 1 to 50 µg/mL for obtaining calibration curves.

Quantitative analysis of phenolics by TQ-S. Quantitative analyses of phenolics were carried out on a Waters AQCUITY UPLC system, equipped with Waters TQ-S triple-quadrupole mass detector (Waters Corp., Milford, MA, USA). The equipment consisted of a quaternary solvent manager, sample manager, column heater, interfaced with a mass spectrometer equipped with an ESI source, operating in negative mode. Instrument control and data processing were performed using MassLynx™ software. Gradient conditions, column parameters and temperature, flow rate and injection volume were the same as described in the section on non-targeted analysis. Nitrogen was used both as drying and nebulizing gas at 1000 and 150 L/h flow for desolvation and at the cone gas, respectively. The desolvation temperature was set at 500 °C. Capillary and cone voltages were 1.8 kV and 25 eV, respectively. The quantification was performed using the external standards (tiliroside, elagic and *p*-coumaric acids, kaempferol-3-glucuronide, quercetin-3-glucuronide, and catechin). Standard stock solutions were prepared in MeOH and subsequently diluted to working concentrations. The amounts of individually identified anthocyanins were expressed as mg/100 g of DW of strawberry pomace.

### 2.7. Statistical Analysis

MS Excel 2016 was used for calculations of mean values and standard deviations. All data were expressed as mean ± standard errors (SD). Significant differences among the means were determined by ANOVA, using the statistical package GraphPad Prism 6.01 software (2012) to identify significant differences.

## 3. Results and Discussion

### 3.1. Composition of the Strawberry Pomace

Dried pomace (all values in g/100 g) contained 5.6 ± 0.2 H_2_O, 12.03 ± 0.4 fat, 5.3 ± 0.1 ash and 13.3 ± 0.3 proteins. Other macrocomponents were not determined; however, they should consist mainly of various carbohydrates. A similar amount of fat (11.6 g/100 g DW of pomace) was reported Pieszka et al. [29]; while Górnaś et al. [30] determined only 3.4 g/100 g DW. The content of fat in pomace largely depends on seed fraction; Sójka et al. [31], reported that 40% of strawberry pomace consisted of seeds; while the content of ash, depending on the harvesting season, was in the range of 4.0–7.6 g/100 g DW. Pomace contamination with sand at different seasons may also have significant effect on the content of ash. In general, chemical composition and quality of dried fruit pomace depend on the raw material, which may vary due to the differences in growing and climatic conditions, fertilization and other agronomic treatments, as well as drying method and its parameters [32].

### 3.2. Characterization of Polar Phytochemicals of Strawberry Pomace Extracts

Phenolic compounds possess protective effect due their antimutagenic, antioxidant, antimicrobial, anticarcinogenic and antiinflammatory effects [33]. The results obtained in our study support the bioactive potential of strawberry pomace extracts. The compounds are listed in Table 1 with the relevant identification data, namely retention time, experimental mass *m*/*z*, molecular formula, and the Q-TOF-MS/MS fragmentation patterns. All compounds were characterized by the interpretation of their mass spectra recorded by Q-TOF-MS and comparing it with the data available in the literature and open databases (Chemspider, MetFusion). When available, the identification was supported by using authentic standards. In total, 15 phenolic metabolites were identified including organic/phenolic acids. The compounds 1, 2, 3, 4, belonging to organic acids were detected in the all analyzed extracts; these compounds co-eluted together and were identified as malic, citric, succinic and quinic acids, which agrees with the previously reported data [30] and open access databases (quantitative analysis is shown in Figure 1). The compounds 5, 6, 10, 11, 13 and 15 were identified as catechin, *p*-coumaric acid, ellagic acid, quercetin-3-glucuronide, kaempferol-3-glucuronide and tiliroside. Their structures were confirmed by authentic standards and fragmentation patterns (Table 1). The compounds 11 and 13 had a pseudomolecular ion [M-H]^−^ at *m*/*z* 477.0674 and 461.0726, fragmentation patterns revealed the presence of a fragments at *m*/*z* 301.0356 and 285.0374, respectively, which indicates the presence of aglycons quercetin and kaempferol, respectively.

The compound 15 was identified as tiliroside, which gave an *m*/*z* 593.1297 (matching molecular formula C_30_H_25_O_13_) and intense fragments at *m*/*z* 447.0932 and *m*/*z* 285.0420. The fragment *m*/*z* 285.0420 corresponds kaempferol molecular ion C_15_H_9_O_6_, which is aglycon of this glycoconjugate, while fragment loss of 308 amu reveals the simultaneous presence of coumaroyl moiety. According to Bergantin et al. [34] the loss of 308 amu is formed by the MS/MS daughter ions of most intense fragmentation mass change, namely *m*/*z* 162 for hexose and *m*/*z* 146 for coumaroyl group. The compound 7 was tentatively identified as apigenin-hexoside [35]. This compound gave molecular ion [M–H]^−^ at *m*/*z* 431.0979 (C_21_H_19_O_10_), MS/MS fragment at 269.0448 (C_15_H_9_O_5_) corresponding fragmentation pattern of apigenin aglycone. The loss of 162 amu corresponds sugar unit of hexoside. The compound 9 gave an ion *m*/*z* 447.0571 (C_20_H_15_O_12_) and intense fragment *m/z* 300.9989 (C_15_H_9_O_7_) corresponding to quercetin aglycone. The former could be the result of a loss of 146 amu indicating [M-146] rhamnose unit. Thus, this compound was tentatively identified as quercetin-rhamnoside. The compound 12 gave a pseudomolecular ion [M–H]^−^ at *m*/*z* 447.0931 corresponding C_21_H_19_O_11_ molecular formula, whereas the released unique MS/MS fragment at *m*/*z* 285.0374 (C_15_H_9_O_6_) indicates the loss of [M-162] hexose unit; this compound was tentatively identified as kaempferol-hexoside.

Individual content of phenolic metabolites. In order to quantify the phenolic metabolites generated in the all obtained extracts, tandem MS/MS was used due to its specificity, sensitivity and selectivity. The generated metabolites were determined in multiple reaction monitoring (MRM) mode. The available standards were used for quantification. The concentrations of the quantified phenolic compounds are presented in Figure 1. Ellagic acid was the main component in the all analyzed samples, varying between 8.7 and 64.1 mg/100 g DW. In general, our results agree with the data of Šaponjac et al. [3], who reported in strawberries approx. 2.7 mg/100 g fresh weight (FW) of ellagic acid. Aaby et al. [4] reported that the content of ellagic acid varied from 0.2 to 87.3 mg/100 g FW in the extracts of strawberry flesh and achenes obtained with acetone and H_2_O. The other predominant compound was quercetin-3-glucuronide, which was found in the highest amounts in PLE-EtOH and PLE-H_2_O extracts, 38.9 and 36.5 mg/100 g DW, respectively. The amount of kaempferol-3-glucuronide and tiliroside was in the range of 0.3–35.1 and 1.9–10.3 mg/100 g DW, respectively. SLE-H_2_O extract had the lowest content of the identified compounds; consequently, their solubility could be influenced by the lower extraction temperature. Previous results on individual phenolic compounds quantified in strawberry pomace extracts showed that quercetin and kaempferol derivatives were the main components, constituting 18.4–37.9 and 10.2–39.7 mg/100 g DW, respectively [31]. The content of some other reported phytochemicals, recovered by the traditional and pressurized extraction methods were lower; e.g., the content of *p*-coumaric acid and catechin in the SLE-H_2_O extract was only 1.9 and 0.1 mg/100 g DW, respectively.

Šaponjac et al. [3] determined higher amount of catechin, which, depending on strawberry species and cultivar, varied from 19.6 to 135.2 mg/100 g FW. It indicates that the variation of polyphenolic composition in strawberry pomace depends on the numerous factors, such as genetic characteristics of fruits (cultivar/genotype), environmental peculiarities (geographical cultivation site, climatic conditions), ripeness and processing aspects.

Qualitative analysis of organic acids. Accumulation of organic acids in the cells of berries is one of the major factors, that play an important role in strawberry quality. It was reported that the degree of ripeness and timing vary greatly among species and varieties and is highly susceptible to the climatic conditions [36]. Sugars and organic acids are important contributors of strawberry taste and flavor [37]. In our study citric and quinic acids were the main organic acids determined in the strawberry pomace extracts (Figure 2). The concentration of citric acid in the all analyzed extracts was from 911.4 to 2585.3 mg/100 g DW; wile the concentration of other organic acids such as malic and succinic was lower; for instance, malic and succinic acids in SLE-EtOH/PLE-EtOH extracts were present at 269.3/407.3 and 23.1/58.7 mg/100 g DW, respectively. The amount of quinic acid in SLE and PLE extracts was from 720.7 to 2417.4 mg/100 g DW. To the best of our knowledge, quinic acid was not previously quantified in the strawberry pomace. In general, the concentrations of organic acids were higher in aqueous extracts both isolated by PLE and SLE methods. Górnaś et al. [30] reported that the content of citric, malic and succinic acids in the dry strawberry pomace was 47, 27, and 13 g/kg DW, respectively. According to Reißner et al. [38] the amount of organic acids in pomace highly depends both on berry composition and the effectiveness of juice extraction, which influences the transfer of acids into the juice.

Quantitative and qualitative analysis of anthocyanins. Anthocyanins are water-soluble pigments, which have been commercially used as antioxidants, nutraceuticals, and red natural colorants in different foodstuffs, cosmetics and medicines. The anthocyanins are responsible for the so-called cyanic colors of numerous plant species and their fruits, ranging from pink to red and from violet to dark blue. The anthocyanins are especially abundant in some dark-colored berries such as bilberries, black currants, and others [38,39,40]. The identification data of anthocyanins are presented in Table 2. The anthocyanins were identified by the interpretation of their mass spectra determined by the Q-TOF-MS, relative intensity of MS/MS fragmentation ions and DAD signals and, finally, by comparing with the literature data. Cyanidin-3-O-glucoside *m*/*z* 449.1071 (C_21_H_21_O_11_) was unambiguously identified by comparing its retention time, UV–Vis spectroscopic data, and the pseudomolecular ion [M]^+^ with an authentic standard. The main anthocyanins, pelargonidin-3-O-gluoside and pelargonidin-3-O-rutinoside, were identified based on their retention times and MS/MS fragments; these compounds gave [M]^+^ at *m*/*z* 433.1132 (C_21_H_21_O_10_) and *m*/*z* 579 (C_27_H_31_O_14_) and the characteristic fragment ion at *m*/*z* 271.0592, with neutral loss of of [M-162] and [M-308] corresponding to hexose (glucose) and rutinose units, respectively.

Acylated anthocyanins such as pelargonidin 3-(6″-malonyl)-glucoside and cyanidin 3(-6″-coumaroyl)-glucoside were identified using the results reported by Šaponjac et al. [3] and Aaby et al. [4]. These compounds exhibited a molecular [M]^+^ ions at *m*/*z* 519.1128 (C_24_H_23_O_13_) and 595.1438 (C_30_H_27_O_13_), the fragment ions at *m*/*z* 271.0592 and 287.0683, which correspond to pelargonidin and cyanidin aglycones, respectively. According to Bergantin et al. [34] glycosylated derivatives were recognized by the most intense MS/MS transition, usually determined by the characteristic neutral loss of glycosyl moiety. Less intense MS/MS peaks can refer to typical fragmentations of involved saccharide unit or the loss of malonyl and coumaroyl groups. MS/MS daughter ions indicating on the simultaneous loss of malonyl or acetyl, coumaroyl and glycosyl moieties as the most intense fragments, a mass change of 248 *m*/*z* (*m*/*z* 162 for hexose and *m*/*z* 86 for malonyl group) and *m*/*z* 308 (i.e., *m*/*z* 162 for hexose and *m*/*z* 146 for coumaroyl group), respectively. The sum of the concentrations of the all identified anthocyanins shows that PLE-EtOH (164.83 mg/100 g DW of pomace) had the highest content of the quantified anthocyanins among other extracts, followed by PLE-H_2_O (126.9 mg/100 g DW), SLE-H_2_O (89.2 mg/100 g DW), and SLE-EtOH (80.2 mg/100 g DW) (Table 2).

The total anthocyanin content was expressed as a cyanidin-3-O-glucoside. Our results agree with the previously reported data [3,4,31], which also reported the presence of pelargonidin-3-O-glucoside, cyanidin-3-O-rutinoside, cyanidin-3-O-glucoside, cyanidin-3-(6″coumaroylglucoside) in different strawberry extracts. Pelargonidin-3-O-glucoside was the major anthocyanin found in the analyzed strawberry extracts constituting approx. 62–47% of the total anthocyanin content, followed by cyanidin-3-(6″coumaroylglucoside), pelargonidin-3-O-rutinoside and cyanidin-3-O-glucoside.

### 3.3. Antioxidant Properties

#### 3.3.1. Antioxidant Characteristics Measured by the Chemical Methods

PLE was selected as an advanced extraction technique, which can quickly and comparatively selectively recover phenolic compounds using food and environmentally safe solvents (GRAS) such as EtOH and H_2_O. The extraction yields, as well as the total amounts of phenolic compounds of the extracts obtained by SLE and PLE are presented in Table 3. The highest yields were recovered by PLE with EtOH and H_2_O, 28.6 ± 1.4 and 24.9 ± 0.6 g/100 g of DW of pomace, respectively. SLE-H_2_O and SLE-EtOH yielded 19.5 ± 1.8 and 14.8 ± 1.3 g/100 g of DW, respectively. PLE extracts also contained higher amounts of phenolic compounds (Figure 1). The extract yield can be influenced by several factors, namely solvent properties (polarity, density), extraction method and time, solvent/solid ratio, temperature, and particle size of the plant material [41]. Furthermore, the viscosity and surface tension of the solvents decrease at higher temperatures, which might increase encroachment of the solvent into the matrix and a faster dissolution. Moreover, mass transfer rate is increased, thus resulting in higher yields in PLE [42,43,44,45]. The distinguishing advantage of PLE vs. SLE is the ability to combine elevated solvent temperature and pressure for achieving fast and efficient extraction within a wide range of compound polarities. In addition, PLE uses 2–4 fold smaller amount of solvent than SLE [41,46,47].

Regarding TPC and antioxidant potential, the efficiency of extraction methods may be evaluated taking into account 2 important indicators: (i) the concentration of the target compounds (or their groups) in the extract, which may be considered as a final or intermediate product for further application; (ii) the level of recovery of such compounds (e.g., antioxidants) from dry plant material mass. The TPC in the extracts varied from 21.5 to 46.8 mg GAE/g and it may be observed that it was significantly higher in the EtOH extract obtained by SLE, while in H_2_O extracts it was similar (Table 3). It may be explained by the remarkably higher yields in PLE-EtOH; in this case, Folin–Ciocalteu reactive substances are diluted with the neutral in this reaction compounds, e.g., pectins or other carbohydrates. It may also be noted that the relationships between TPC values and the amounts of chromatographically quantified flavonoids and phenolic acids do not exist. It should be noted that some other substances, e.g., such as reducing sugars, which are not taken into account in this assay, may interfere in the reaction together with polyphenolics. Aaby et al. [4] reported that the TPC in the H_2_O extracts of strawberry industrial waste was in the range of 21–120 mg GAE/100 g FW.

The solid material after various steps of extractions may still contain bound and not recovered phytochemicals. Therefore, antioxidant properties of non-soluble pomace fractions (Table 3) were monitored after each step of extraction by employing the so-called QUENCHER approach, which has been adapted to the common antioxidant assays [25]. The QUENCHER (QUick, Easy, New, CHEap and Reproducible) approach was proposed for direct assessment of antioxidant capacity of solid material, which is assayed without applying extraction step. It is considered that the reaction between free radical and present in the solid matrix antioxidants can occur at the interface when they are in contact, regardless of the hydrophobicity of the compound of interest [48]. Considering that the TPC in the raw pomace (starting material) was 18.8 mg/g DW, it may be assumed that comparatively small part of antioxidants was recovered from the strawberry pomace by SLE with EtOH and H_2_O: the residual TPC values were 10.2 and 14.3 mg GAE/g DW, respectively. PLE-EtOH and PLE-H_2_O recovered remarkably higher amounts of phenolics; the residual values after extractions were 6.5 mg and 3.0 mg GAE/g DW, respectively. The high TPC in the strawberry pomace suggests that they might be a good source of phytochemicals for value-added products.

DPPH^•^, ABTS^•^^+^ scavenging and ORAC assays were used to evaluate antioxidant potential of strawberry pomace extracts (Table 1). DPPH^•^ scavenging values were remarkably lower compared with ABTS^•^^+^ scavenging values and, although the basic principle of these two methods is somewhat similar, the differences may be explained by the peculiarities of the reaction mechanisms and different reaction medium polarity. Among the analyzed extracts, PLE-EtOH extract demonstrated the highest activity, followed by the PLE-H_2_O and SLE-EtOH extracts: their antioxidant capacity values in the ABTS^•+^, DPPH^•^ and ORAC assays were 148.5–391.9; 28-3–117,2 and 95.4 to 308,9 mg TE/g DW of strawberry pomace, respectively. Aaby and co-authors [4] reported significantly lower antioxidant values for acetone and water SLE extracts of strawberries; e.g., ORAC was 4.1–174 μmol TE/g FW, which is equivalent to 1.02–43.5 mg of trolox.

Strong positive linear correlations were observed between DPPH^•^ scavenging values and the amounts of catechin, *p*-coumaric acid and quercetin-3-glucuronide, which suggests that these compounds may have an important impact on the results of this antioxidant capacity assay. ABTS^•+^ scavenging values are in strong correlation with the amounts of all chromatographically determined phenolic compounds, except for tiliroside. On the other hand, the content of tiliroside correlated with ORAC values, which also strongly correlated with the amounts of phenolic acids, total flavonoids and total phenolic acids. It may be noted that ORAC assay is based on different mechanism (scavenging of peroxyl radicals), which is more relevant to the oxidation processes taking place in the biological systems [24].

It should also be noted that the evaluation of antioxidant capacity of extracts is usually less accurate when the extracted substance contains amino acids, fibers and/or uronic acids [25]. It is evident that regardless the properties of extraction solvent, solid extraction residues do still contain insoluble compounds, which may demonstrate antioxidant properties. This phenomenon was also proved in ABTS^•+^ scavenging assay by applying the QUENCHER method (Table 1) to the strawberry pomace residues after SLE and PLE extractions. The ABTS^•+^ values determined for the solid residues after extractions decreased approx. 2.9–6.4-fold. From this point of view PLE was more efficient, as a significantly lower amount of bioactive compounds remained in the solid fraction after extraction. To the best of our knowledge QUENCHER method has not been previously applied to the strawberry solids at various processing steps.

#### 3.3.2. Evaluation of Cellular Antioxidant Activity (CCA)

CAA was assessed in order to extend the information about antioxidant potential of strawberry pomace extracts. The CAA assay is a more biologically relevant method than the popular chemical antioxidant capacity assays because it accounts for the factors such as uptake, metabolism, and localization of antioxidant compounds within the cells [28]. The presented in the Section 3.1 and Section 3.3.1 results on phytochemical composition and antioxidant capacity of extracts, demonstrated that they depend on the method of extraction and solvent polarity. PLE extracts demonstrated the strongest antioxidant activities in the DPPH^•^, ABTS^•^^+^ scavenging and ORAC assays (only TPC was higher in SLE-EtOH extract). So far as the concentration of individual compounds such as phenolic and organic acids, flavonoids were also remarkably higher in the PLE extracts they were selected for further evaluation. Both PLE extracts showed strong antioxidant activity with CAA values of 11.17 ± 1.88 and 5.9 ± 2.9 μmol QE/mg of dry weight for H_2_O and EtOH extracts, respectively, while the EC_50_ values were 0.24 ± 0.01 and 0.50 ± 0.3 mg/mL, respectively (data not shown, the values were obtained together with quercetin EC_50_ and used to calculate CAA value). Wolfe et al. [49] reported CAA of 42.2 ± 3.3 μmol QE/100 g fresh strawberries (in case of using HepG2 cells and applying PBS wash), while EC_50_ value of the acetone extract diluted with methanol was 11.8 ± 0.9 mg/mL. To the best of our knowledge CAA for strawberry pomace extracts has not beed reported previously, while antioxidant potential of strawberry and its pomace extracts was studied previously by using various chemical methods, mainly ABTS^•^^+^, DPPH^•^, ORAC, and FRAP [3,29,50,51].

### 3.4. Antiproliferative Effects and Cytotoxicity of Strawberry Pomace Extracts

In order to evaluate the antiproliferative activity, HT29 cells were used at the exponential grow phase, while cytotoxicity was assessed using Caco-2 cell line as the best accepted intestinal model. The extracts were assessed at their maximum solvent concentration allowed to use where EtOH extract showed capability of inhibiting cancer cell grow with an EC_50_ value of 5.1 ± 0.2 mg/mL (Figure 3A), without compromising normal epithelia once no cytotoxicity was observed on Caco-2 (Figure 3B). Water extract was tested at its maximum concentration allowed showing no cytotoxicity or antiproliferative effect.

Berry extracts have been widely explored due to high polyphenol content, which provides a strong antioxidant capacity to the extracts. Especially strawberries have been described as a good source of anthocyanins, flavonols, tannins and hydroxycinnamic acids which are already known to have strong bioactivity such as antioxidant, antiproliferative and cardiovascular potential [52,53,54,55]. High concentration of individual phytochemicals and strong antioxidant capacity may be responsible for the inhibitory effects of PLE-EtOH and PLE-H_2_O extracts against cancer cell growth (Figure 1, Table 3). For instance, well-known anticancer compounds ellagic acid [56] and quercetin-3-glucuronide, which were abundant in EtOH extract, may play an important role for antiproliferative activity. However, due to the compositional complexity of secondary plant metabolites, antioxidant capacity values measured by the in vitro methods do not always correlate with their activities in vivo. Amatori et al. [55] reported the antiproliferative effect of methanolic polyphenol-rich strawberry extract on breast cancer using MCF-7 and A17 cell lines: the inhibition of cancer cell growth was time- and dose-dependent. Moreover, EC_50_ for A17 cell line determined after 24 h of incubation (1.14 ± 0.29 mg/mL) was non-cytotoxic in normal breast and fibroblast cells WI38 and NIH-3T3. The antiproliferative effect on HT29 cell line has been also tested and it was shown that strawberry extracts inhibit cell growth in a dose-dependent manner [54]. These results are in accordance with the results obtained in our study for EtOH extract of strawberry pomace: it showed antiproliferative activity against HT29 cells at the highest concentration. H_2_O extract was not active; most likely due to the higher content of secondary metabolites with anticarcinogenic properties in the EtOH extract. In addition, extraction temperature in case of PLE-EtOH was remarkably lower (90 °C), than in case of PLE-H_2_O (110 °C), and this factor may have an effect on the compounds stability, especially anthocyanins, which at higher temperature may loose sugar moiety providing a protective effect for unstable anthocyanidins. Anthocyanins, ellagic and coumaric acids, flavonoids (catechin, kampferol-3-glucuronide and tiliroside) were the major constituents in EtOH extract (Section 3.2). As mentioned before these compounds have been reported to have strong antiproliferative activities against various cancer cells and therefore are considered as promising preventing means against cancer.

## 4. Conclusions

Strawberry pomace was valorized as a source of phenolic compounds, organic acids, flavonoids, and anthocyanins by comparing conventional solid-liquid (SLE) and pressurized liquid (PLE) extractions with ethanol EtOH and H_2_O. Cytotoxicity, antiproliferative and cellular antioxidant activity of strawberry pomace extracts isolated by PLE is reported for the first time. PLE-EtOH extract showed the highest antiproliferative activity with no cytotoxicity associated. Both PLE extracts demonstrated strong antioxidant potential and protected Caco-2 cells upon stress stimuli. It may be assumed that the biological activities of the extracts are due to the presence of the identified and quantified in this study flavonoids (anthocyanins, catechin) and phenolic acids (ellagic, coumaric). Based on these findings PLE extracts may be regarded as promising antiproliferative and antioxidant substances in disease prevention. Therefore, they can find wide applications in developing new functional foods and nutraceuticals with health-beneficial properties. Future studies should focus on the use of strawberry pomace extracts in the selected food products and evaluation of their effects on the quality, while the studies on their health benefits should be continued by the in vivo assays.

## Figures and Tables

**Figure 1 foods-10-01780-f001:**
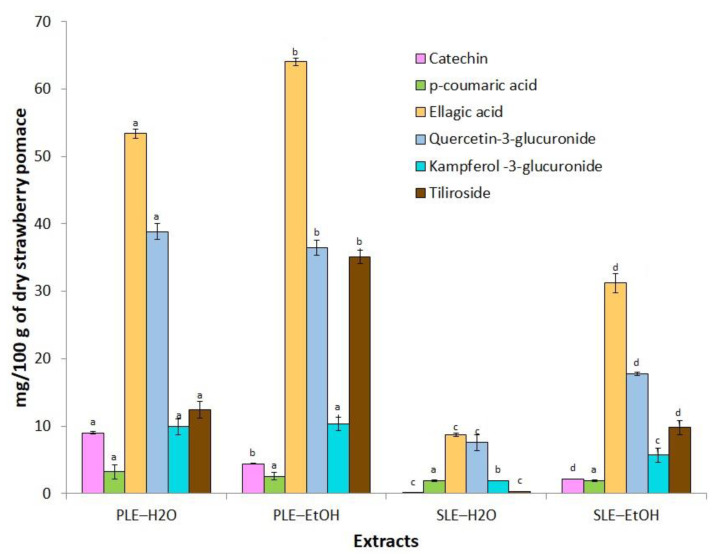
The content of flavonoids and phenolic acids in strawberry pomace extracts obtained by pressurized liquid (PLE) and solid liquid extraction (SLE) with different solvents: expressed as a mean with standard deviation (*n* = 3); different letters (a–d) indicate significant differences determined by ANOVA Duncan’s test (*p*
*<* 0.05).

**Figure 2 foods-10-01780-f002:**
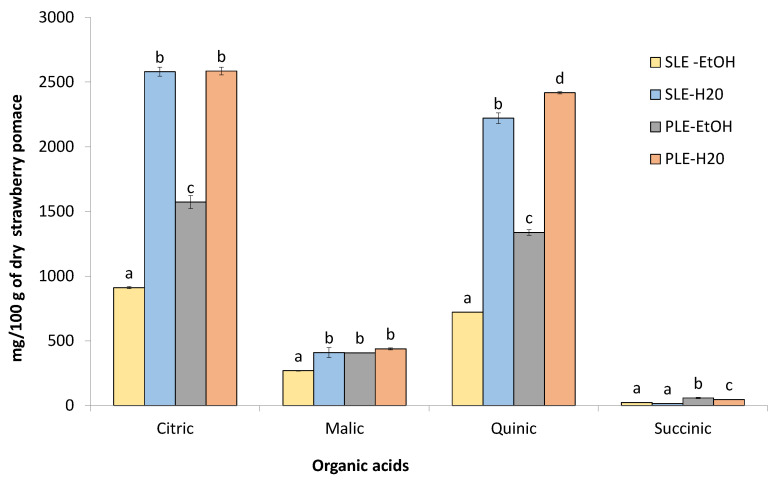
Concentration of organic acids in SLE and PLE extracts of strawberry pomace: the values expressed as means with standard deviations (*n* = 3); different letters (a–d) indicate significant differences determined by ANOVA Duncan’s test (*p* < 0.05).

**Figure 3 foods-10-01780-f003:**
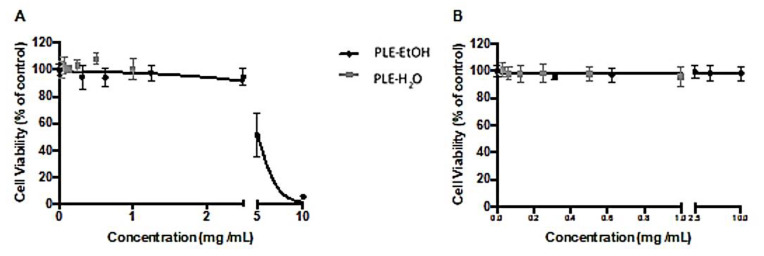
Dose-response curves of strawberry extracts. Antiproliferative (**A**) and cytotoxicity (**B**) effects using HT29 and Caco-2 cell lines, respectively. Results are expressed in terms of mean ± SD performed in triplicate.

**Table 1 foods-10-01780-t001:** The main constituents of strawberry pomace extracts identified by the non-targeted UPLC-Q-TOF analysis: SLE—solid-liquid extraction; PLE—pressurized liquid extraction.

No.	RT (min)	Precursor Ion [M–H]^-^	Ion Formula	Negative Ions MS^2^ *m*/*z* (% in MS)	Structural Assignment	Extracts			
						SLE- EtOH	SLE- H_2_O	PLE- EtOH	PLE- H_2_O
1	0.5	133.0131 ^a^	C_4_H_5_O_5_	-	Malic acid	+	+	+	+
2	0.59	191.0186 ^a^	C_6_H_7_O_7_	-	Citric acid	+	+	+	+
3	0.65	117.0182 ^a^	C_4_H_5_O_4_	-	Succinic acid	+	+	+	+
4	0.72	191.0550	C_7_H_11_O_6_	-	Quinic acid	+	+	+	+
5	2.35	289.0716 ^a^	C_15_H_13_O_6_	-	Catechin	+	-	+	+
6	2.55	325.0925 ^b,c^	C_15_H_17_O_8_	145.0291 (50.3)	*p*-coumaric acid hexoside	+	-	+	+
7	2.95	431.0979 ^b,c^	C_21_H_19_O_10_	147.0075 (4.7) 241.0498 (10.2) 269.0448 (100)	Apigenin-7-glucoside	+	-	+	-
8	3.65	163.0402 ^a^	C_9_H_7_O_3_	-	*p*-coumaric acid	+	+	+	+
9	3.95	447.0571 ^b,c^	C_20_H_15_O_12_	300.9986 (100)	Quercetin-3-rhamnoside	+	+	+	+
10	4.15	300.9987 ^a^	C_14_H_5_O_8_	-	Ellagic acid	+	+	+	+
11	4.35	477.0674 ^a,b^	C_14_H_17_O_13_	301.0356 (100) 151.0035 (8.4)	Quercetin-3-glucuronide	+	+	+	+
12	4.75	447.0931 ^b,c^	C_21_H_19_O_11_	285.0374 (100)	Kaempferol-hexoside	+	+	+	+
13	4.85	461.07726 ^a,b^	C_21_H_17_O_12_	285.0372 (100)	Kaempferol-3-glucuronide	+	+	+	+
14	5.1	137.0243 ^b,c^	C_7_H_5_O_3_	-	Hydroxybenzoic acid	+	+	+	+
15	5.6	593.1297 ^a,b^	C_30_H2_5_O_13_	145.0303 (9.3) 285.0420 (100) 447.0932 (49.5)	Tiliroside	+	+	+	+

^a^ Confirmed by a standard; ^b^ Confirmed by comparison with the literature data; ^c^ Confirmed by parent ion mass using open access chemical databases (Chemspider, MetFusion).

**Table 2 foods-10-01780-t002:** Anthocyanin’s profile in strawberry pomace extracts.

RT, min	Structure Assignment	Precursor Ion [M]^+^ (*m/z*)	Molecular Formula	SLE- H_2_O	SLE-H_2_O	PLE- EtOH	PLE-H_2_O
				mg/100 g DW of Strawberry Pomace			
5.5	Cyanidin-3- glucoside *	449.1071	C_21_H_21_O_11_	8.7 ± 0.1 ^a^	16.1 ± 0.4 ^b^	16.9 ± 0.4 ^c^	14.4 ± 0.1 ^d^
6.2	Pelargonidin-3-glucoside **	433.1132	C_21_H_21_O_10_	49.1 ± 0.7 ^a^	55.5 ± 0.1 ^b^	78.9 ± 0.9 ^c^	60.1 ± 1.3 ^d^
6.4	Pelargonidin-3-rutinoside **	579.1696	C_27_H_31_O_14_	9.1 ± 0.1 ^a^	12.6 ± 0.7 ^b^	19.9 ± 0.1 ^c^	14.8 ± 0.2 ^d^
7.6	Pelargonidin-3-(malonyl)- glucoside **	519.1128	C_24_H_23_O_13_	-	-	20.1 ± 0.7 ^a^	16.6 ± 0.1 ^a^
8.1	Cyanidin-3- (coumaroyl)- glucoside **	595.1438	C_30_H_27_O_13_	13.3 ± 0.1 ^a^	-	31.9 ± 0.8 ^b^	20.4 ± 0.8 ^c^
	Total			80.2	84.2	167.7	126.3

* Confirmed by a standard; ** Confirmed by parent ion mass using comparing with the literature data. Values expressed as a mean and standard deviation (*n* = 3); different superscript letters (a–d) indicate significant differences determined by ANOVA Duncan’s test (*p* < 0.05).

**Table 3 foods-10-01780-t003:** Total phenolic content and antioxidant capacity of extracts and solid material.

Extract	Yield g/100 g DW	TPC		DPPH^•^	ABTS^•^^+^		ORAC
		mg GAE/g Extract	mg GAE/g DW of Strawberry Pomace (Quencher Approach)	mg TE/g Extract	mg TE/g Extract	mg TE/g DW of Strawberry Pomace (Quencher Approach)	mg TE/g Extract
SLE-EtOH	14.7 ± 1.3	46.8 ± 1.8	10.2 ± 0.5	57.1 ± 2.8	180.2 ± 4.2	29.3 ± 0.9	201.8 ± 2.6
SLE-H_2_O	19.5 ± 1.8	30.3 ± 0.9	14.3 ± 0.8	28.3 ± 1.2	148.5 ± 1.1	42.8 ± 0.7	95.4 ± 0.5
PLE-EtOH	28.6 ± 1.4	21.5 ± 0.9	6.5 ± 0.4	65.3 ± 3.13	291.6 ± 3.8	19.6 ± 0.4	308.9 ± 1.4
PLE-H_2_O	24.9 ± 0.6	29.6 ± 1.1	3.0 ± 0.2	117.2 ± 2.7	391.95 ± 2.1	21.5 ± 0.2	222.3 ± 3.5
Dried pomace	-	-	18.8 ± 1.2	-	-	125.4 ± 3.7	-

Values expressed are means of 5 replicates with standard deviations ± S.D of three parallel measurements.

## Data Availability

Data is available from the authors.

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
