# Peer review of "Recovery of Bioactive Compounds from Strawberry (Fragaria × ananassa) Pomace by Conventional and Pressurized Liquid Extraction and Assessment Their Bioactivity in Human Cell Cultures"

_foods, 2021, doi:10.3390/foods10081780_

Round 1
Reviewer 1 Report
This is a manuscript of high interest for Foods since the objectives of the work are relevant and results and conclusions of great value have been obtained.
However, I consider that the manuscript lacks some issues that must be addressed. Below I list these questions:
Introduction:
The taxon under study must be written with the author. A correct and complete identification must be made. To understand the importance of the study, the introduction should deal with the genus Fragaria in general, the species it contains and the origin of the taxon under study, since we are talking about a hybrid according to the authors of the manuscript.
Only by having a clear description of the genus and the taxon under study will we be able to understand the manuscript.
More information about what is known about the taxon under study should also appear in the introduction. Works carried out on it or on nearby species, industrial importance, place of cultivation, chemical composition, known biological activities of its chemical components and extracts, etc….
Material and method:
It is quite complete and clear. However, section 2.2 deals with the material studied, where there is no herbarium control of the sample studied. A herbarium testimony is essential in this type of work to be able to replicate the experiments. At least, if there is no herbarium material, the brand or company that has done the cultivation could be included.
Results and Discussion
They are correct, but if the comments made in the introduction are taken into account, I think that the discussion of the results will be more valuable.
Other minor comments:
There is a layout problem in figure 1 and table 2. They are cut off by staying between two sheets.
Author Response
please find our response to reviewer in the attachment

Reviewer 2 Report
In general, the discussion of the obtained data should be better explored. It lacks a correlation between the phytochemical composition of the extracts and the obtained bioactivities, to improve the quality of the manuscript.
Line 24: I suggest to eliminate or reformulate this sentence.
Line 39: Organic residues such as strawberry pomace are not an environmental pollution problem since they do not contaminate soil or water.
Line 52: Please reformulate this sentence.
Line 57: in vivo (italic)
Lin 193: The authors do not explain why they use this new QUENCHERS procedure. Please explain the procedure. What are the advantages?
Line 240: Phytochemicals were identified…
Line 350: strawberry pomace suggests that they….
Line 354: Makes no sense to compare the values between DPPH and ABTS. These are different assays with different mechanisms of action, so it is expected that the values are different.
Lin 375: Why only the PLE extracts were evaluated for the cellular assays? Please explain throughout the manuscript it is not clear.
Line 421: This sentence should take into account the results obtained by HPLC. As I said before, I suggest to put the bioactivity assays after the HPLC data and then the discussion of the obtained results should be based on the phytochemical composition of the extracts. Also, correlation studies between HPLC data (individual and the sum of the phytochemicals contents) and bioactivities are missing and should be done to improve the discussion of results.
Figure 2 – Please change the color of the bars to become clearer to the reader. Also, superscript letters are missing. Also, I suggest to make a graphic with the sum of the amounts of flavonoids and phenolic acids. The same comments apply to Figure 3.
Author Response

(The authors gave the same response as above.)

Round 2
Reviewer 1 Report
The authors have clearly responded to each of my comments and have resolved some doubts that I had about the manuscript.
In addition, the authors have improved the manuscript with respect to the initial version.
In relation to my comment on the material and method, I consider that the information provided by the authors should appear in the text and thus give more scientific weight to the work.
I refer to this text: "In addition, the dried sample of strawberry pomace, which was used in this study, is stored at 4 °C in the Department of Food Science and Technology (no. SP-2017). It is available upon request " .
Author Response
We are very grateful the Reviewer for positive evaluation. We have added the requested information into the revised manuscript (highlighted).
Reviewer 2 Report
The authors have clearly responded to each of my comments and have resolved some doubts that I had about the manuscript. In addition, the authors have improved the manuscript with respect to the initial version. Some minor comments:
I suggest to add a table with the correlations obtained from the polyphenolic contents (phenolic acids and anthocyanins) and antioxidant activities (it could be as supplementary material). There was no correlation with the anthocyanins content? The authors do not mention it. Also, the authors do not mention the correlation for the CAA and the polyphenolic content. What compounds contributed the most?
Some references are missing (error). Please correct it.
Table 2 – I suggest to add the sum of anthocyanins at the end of the Table.
Line 656 – Both PLE extracts….
Line 665 – in vivo (italic).
Author Response
Comment: The authors have clearly responded to each of my comments and have resolved some doubts that I had about the manuscript. In addition, the authors have improved the manuscript with respect to the initial version. Some minor comments:
Response: We are very grateful the Reviewer for positive evaluation. We have added the requested information into the revised manuscript
Comment: I suggest to add a table with the correlations obtained from the polyphenolic contents (phenolic acids and anthocyanins) and antioxidant activities (it could be as supplementary material). There was no correlation with the anthocyanins content? The authors do not mention it. Also, the authors do not mention the correlation for the CAA and the polyphenolic content. What compounds contributed the most?
Response: The correlations may be calculated; however, we believe that their value for the reader would not be high. The principles of spectrophotometric assays of antioxidative activity and chromatographic quantification of individual compounds are completely different; therefore, correlation between these data can be considered as rather indicative (sometimes even controversial). In addition, such calculations would require quite a lot of technical work, while the added value would be negligible. Therefore, we believe that short comments about positive correlations between the selected groups of compounds and antioxidative capacity value would be sufficient at this stage of work.
Comment: Some references are missing (error). Please correct it.
Response: We have checked the citations and the list of references and have corrected found mistakes and inaccuracies.
Comment: Table 2 – I suggest to add the sum of anthocyanins at the end of the Table.
Response: Table 2 has been extended by adding the row with the total anthocyanin concentration
Comment: Line 656 – Both PLE extracts….
Response: Corrected
Comment: Line 665 – in vivo (italic).
Response: Corrected